# Few-shot Visual Reasoning with Meta-analogical Contrastive Learning

**Youngsung Kim**[1]    **Jinwoo Shin**[2]    **Eunho Yang**[2]    **Sung Ju Hwang**[2]
[1]Samsung Advanced Institute of Technology, Samsung Electronics,
[2]Korea Advanced Institute of Science and Technology (KAIST)
[1]yskim.ee@gmail.com, [2]{jinwoos,eunhoy,sjhwang82}@kaist.ac.kr

## Abstract

While humans can solve a visual puzzle that requires logical reasoning by observing only few samples, it would require training over a large number of samples for state-of-the-art deep reasoning models to obtain similar performance on the same task. In this work, we propose to solve such a few-shot (or low-shot) abstract visual reasoning problem by resorting to *analogical reasoning*, which is a unique human ability to identify structural or relational similarity between two sets. Specifically, we construct analogical and non-analogical training pairs of two different problem instances, e.g., the latter is created by perturbing or shuffling the original (former) problem. Then, we extract the structural relations among elements in both domains in a pair by enforcing analogical ones to be as similar as possible, while minimizing similarities between non-analogical ones. This analogical contrastive learning allows to effectively learn the relational representations of given abstract reasoning tasks. We validate our method on RAVEN dataset, on which it outperforms state-of-the-art method, with larger gains when the training data is scarce. We further meta-learn our analogical contrastive learning model over the same tasks with diverse attributes, and show that it generalizes to the same visual reasoning problem with unseen attributes.

## 1 Introduction

The *visual reasoning* task proposed in recent works [1, 2] often involves visual puzzles such as *Raven Progressive Matrices (RPM)*, whose goal is to find an implicit rule among the given image panels, and predict the correct image panel that will complete the puzzle (see Figure 1b). Since one should identify a common relational similarity among the visual instances with diverse attributes (shape, size, and color), solving such a visual reasoning problem requires reasoning skills which might help take a step further toward general artificial intelligence.

Recently, researchers in the machine learning community have proposed specific models for visual reasoning using deep learning [1, 3, 4, 5]. Deep neural networks are powerful and effective predictors for a wide spectrum of tasks such as classification and language modelling, and has yielded impressive performance on them. However, while humans can solve these logical and abstract reasoning problems by observing only few samples, the state-of-the-art deep reasoning models still require large number of training samples to achieve similar performance on the same task (see Figure 1c).

We hypothesize that such sample-efficiency comes from the flexibility of human intelligence, which can generalize well across different problems by identifying the same pattern in the two problem-pairs with *analogical reasoning*. For instance, given any two pairs of relationships, "$A$ is to $B$" and "$C$ is to $D$", one can form an analogy from the morphological parallelism between the two, e.g., as "$A$ is to $B$ as $C$ is to $D$" or "$A : B :: C : D$" (Figure 1a). With such an analogical relationship established between the pairs, one will be able to predict any single entity given the rest of the entities. For visual

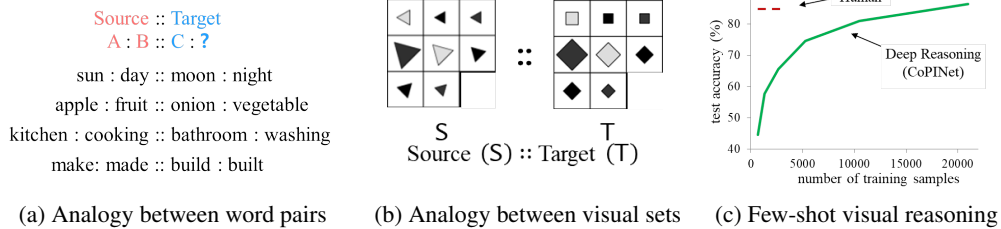

| (a) Analogy between word pairs | (b) Analogy between visual sets | (c) Few-shot visual reasoning |

Figure 1: **Concept.** Analogy defines the relational or structural similarity between two sets. Analogy can be defined between (a) a pair of words or (b) between sets of visual elements. Analogies allow for sample-efficient learning, which is not possible with (c) the state-of-the-art deep abstract visual reasoning network [3] which suffers from dramatic performance degeneration with fewer number of training examples.

reasoning problems, we could also identify a visuospatial analogy [6] between two *sets* of visual entities. In general, any pair of sets from which we could identify certain relational or structural similarity between the two sets can suffice as an analogy. *Analogical reasoning* has been extensively investigated to understand human reasoning process in the field of cognitive science [7] and artificial ingelligence [8, 9].

However, the existing deep visual reasoning models do not have ability to perform such analogical reasoning. To overcome this limitation, we propose a deep representation learning method that exploits such analogical relationship between two sets of instances, with the special focus on the abstract visual reasoning problem. This will allow the model to learn representations that capture relational similarities between the entities, in a sample-efficient manner. Specifically, we train an end-to-end abstract visual reasoning network to learn the abstract concepts by learning an analogy between two problems with the same implicit rules among the set elements, but with different individual elements (Figure 1b). The set that is paired with the original problem in an analogy can be either constructed by perturbing the original problem, or by generating a problem with different attributes. Since we know that the abstract relationships between the elements in two sets are the same, their relational representations should be as similar as possible. On the other hand, the similarity should be minimized between a pair of problems with different relations among the set elements, even when the two are identical at the element level.

To this end, we propose a contrastive learning [10, 11] framework to maximize the similarity between analogical pairs of problems while minimizing the similarity across non-analogical pairs. which we refer to as *Analogical Contrastive Learning* (ACL). We further propose a meta-learning algorithm to train the model over analogical pairs with different attributes, such that it can generalize to unseen attributes (Meta-ACL). We validate our methods on conventional many-shot visual reasoning tasks, as well as few-shot visual reasoning tasks with both seen and unseen attributes, on which it significantly outperforms existing methods. The summary of our contribution is as follows:

- We tackle a challenging problem of *few-shot visual reasoning*, whose goal is to learn to perform abstract visual reasoning with only a few-instances of the given problem, while generalizing to problems with unseen attributes.

- We propose a simple yet novel *meta-analogical contrastive learning* framework that allows the model to capture the relational similarity between a pair of problems.

- We validate our model on a visual reasoning benchmark, RAVEN, and show that it significantly outperforms existing methods under both the conventional and few-shot learning setting, as well as on generalization to problems with unseen attributes.

## 2   Related Work

**Analogy and analogical reasoning.** *Analogies*, which are defined on a pair of sets with the same structures or relation among the elements, are widely used as important heuristics to make new discoveries [12], as they help obtain new insights and formulate solutions via analogical reasoning. *Analogical reasoning*, which is a way to reason about a new problem by drawing an analogy between it and an existing problem, has been considered as a unique trait of human intelligence and a key component in building an AI system with general intelligence. In general, if there are common

characteristics between two objects, an analogical argument can be used as follows [12]: "1) S is similar to T in certain (known) respects. 2) S has some further feature Z. 3) Therefore, T also has the feature Z, or some feature Z* similar to Z. 1) and 2) are premises. 3) is the conclusion of the argument." An analogy is a one-to-one mapping between S and T regarding objects, properties, relations and functions [12]. Not all of the items in S are required to correspond to that of T. Hence, in practice, the analogy only identifies selective shared items.

**Visual analogy learning.** Few existing works tackle the problem of analogical learning in the computer vision domain. Hwang *et al*. [13] proposed to encode analogical relationships between object classes as parallelogram constraints in the learned embedding space. Specifically, they enforced the difference of the embeddings of two pairs of classes to be the same. Reed *et al*. [14] further exploited the analogy to generate analogical images, by adding the difference in the feature vector for one pair of images to the feature of another image. Our method shares the high-level motivation of enforcing relational similarities in the learned representation space with these works, while our work is different from theirs in that we propose a constrastive learning framework for learning the analogical relation between a pair of abstract reasoning problems.

**Visual abstract reasoning.** Abstract reasoning aims to understand symbolic representation and relational structure which are found by disentangling behaviors and combinational attributes such as shape, color, and line) from the given image. Disentangled representation for abstract reasoning has also been investigated in [4], where the authors investigated if a disentangled representation captures the salient factors of variations in the sample space. Zheng *et al*. [15] proposes a robust abstract reasoning method, by combining two learning schemes as a teacher and a student model; reinforcement learning is used to optimize strategy to find abstract features learned by the student model where their results are calculated for a reward of the teacher model. In the student model, called logic embedding network, exhaustive embedding learning is conducted using multiple combination of embedding vectors. Representative example of abstract reasoning is also found in alogrithms [1, 5, 16] for the RPM problem. Perhaps the most relevant work to ours is Hill *et al*. [5], which proposes to perform analogical reasoning task on a visual reasoning task, but they focus on transferring a learned relation from one problem domain to another, while we focus on learning relational representations by explicitly enforcing relational similarities between pairs with known analogical relations.

## 3  Visual reasoning with Raven Progressive Matrices

The Raven Progressive Matrices (RPM) [6, 17] is an abstract visual reasoning task, whose goal is to predict the missing visual panel given multiple context panels. Formally, given an *ordered* set of context visual panels $\mathcal{X} = \{\mathbf{x}_i\}_{i=1}^{8}$ (where $\mathbf{x}_i \in \mathbb{R}^{h \times w}$ denotes an image panel) arranged spatially in a 3×3 matrix $\begin{pmatrix} \mathbf{x}_1 & \mathbf{x}_2 & \mathbf{x}_3 \\ \mathbf{x}_4 & \mathbf{x}_5 & \mathbf{x}_6 \\ \mathbf{x}_7 & \mathbf{x}_8 & ? \end{pmatrix}$ with a missing cell at the bottom-right corner (marked by '?'), the goal of this classification task is to predict the index of the correct panel (that should be placed at the missing cell) from the candidate panels (see [1, 2, 3] for more details). The candidate panels ($\mathbf{c}_y \in \mathbb{R}^{h \times w}$) constitutes a choice set $\mathcal{C} = \{\mathbf{c}_y\}_{y=1}^{8}$. For example, one can generate a simple RPM task using a mathematical operation or function $f : \mathbb{R}^{h \times w} \rightarrow \mathbb{R}^{h \times w}$ as follows:

$$\begin{pmatrix} \mathbf{x}_1 & \mathbf{x}_2 = f(\mathbf{x}_1) & \mathbf{x}_3 = f(\mathbf{x}_2) \\ \mathbf{x}_4 & \mathbf{x}_5 = f(\mathbf{x}_4) & \mathbf{x}_6 = f(\mathbf{x}_5) \\ \mathbf{x}_7 & \mathbf{x}_8 = f(\mathbf{x}_7) & ? \end{pmatrix},$$

i.e., the function is applied iteratively in a column-wise manner. In the above example, the missing panel should be $f(\mathbf{x}_8)$. Similarly, one can also generate an RPM task by applying the function in a row-wise manner, e.g., $\mathbf{x}_4 = f(\mathbf{x}_1)$. That is, in order to solve the visual reasoning task, we should infer the hidden rule $f$ (in addition to whether it is applied in column-wise or row-wise manners in this particular example). However, RPM problems in general can be generated with a variety of rules and attributes [1, 3] which cannot be clearly defined in such a mathematical form. The rules and attributes applied in the problem indicate a type of the problem, namely a *task*.

Recent deep reasoning models for RPM focus on measuring relationships between the context and answer panels by extracting exhaustive embedding vectors and then calculating the output score for the embeddings of each context-answer pair. Then, they select the final answer based on the answer panel with the highest relation score. Specifically, Relation Network [16] and Wild Relation

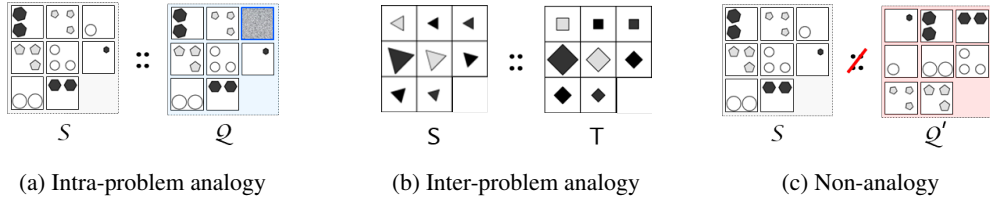

|  (a) Intra-problem analogy | (b) Inter-problem analogy | (c) Non-analogy |

Figure 2: **Different types of analogies in the RPM problem. (a) Intra-problem analogy.** We can create an analogy of the form $\mathcal{X} :: \mathcal{X}'$ by perturbing the original problem with augmentations that do not affect its problem semantics (e.g., by perturbing a random panel). **(b) Inter-problem analogy.** We can also construct an analogy by generating two problems with the same problem semantics but with different attributes (e.g., shapes and colors). **(c) Non-analogy.** We also construct non-analogies as negative pairs for contrastive learning, by randomly shuffling the elements of the original problem instance to break its relational structure.

Network (WReN) [1] calculate all possible relations using a number of combinations of context panels to extract relational representations. CoPINet [3] on the other hand learns to identify the correct answer by maximizing the similarity between the aggregated context panel embedding and the correct answer embedding, while minimizing the similarity between that aggregated context embedding and incorrect answer embeddings.

As mentioned in the introduction, most existing works targeting RPM problems leverage a large number of training samples (e.g., more than 42,000 problem sets for RAVEN (Relational and Analogical Visual rEasonNing) dataset [2] and 1 million sets for PGM (Procedurally Generated Matrices) [5]), and fail to generalize well when trained with few training instances. Moreover, they only consider generalization to the problems with the same attributes (e.g. shapes and colors), and may not generalize well to problems with unseen attributes. In the next section, we introduce a visual reasoning framework which can generalize well even under such challenging scenarios.

## 4   Abstract Visual Reasoning with Meta-analogical Contrastive Learning

A reason why existing models for RPM have poor generalization performance, is because they exploit very little information from each given task itself, by only learning the mapping from each input to its assigned label. To address this limitation, we extract relational similarities by contrasting a pair of problems that constitutes an analogy. Such analogy learning may enable the model to learn with fewer training instances, and generalize better to problems with completely unseen attributes. To this end, we introduce analogies we can exploit from the RPM problems, and propose the analogy learning methods to enforce them in the representation learning, in the following subsections.

### 4.1   Types of analogies in the RPM problem

As explained in Section 3, the panels in each context set $\mathcal{X} \in \mathbb{R}^{8 \times h \times w}$ of an RPM task have an unknown relationships among them (e.g., between rows and columns of the set). That is, there exists an *implicit* relation between some two panels (or two subsets of panels) $a$ and $b$, '$a : b$', where $a, b \in \mathcal{X}$. Our goal is to capture such an implicit relations among the panels by drawing an analogy between two problem instances that have the same relational structure. We exploit two types of analogies: 1) Intra-problem analogies, $\mathcal{X} :: \mathcal{X}'$, that can be found between a context set $\mathcal{X}$ and its perturbation $\mathcal{X}$, and 2) Inter-problem analogies, $\mathcal{X}_\mathsf{S} :: \mathcal{X}_\mathsf{T}$, that exist between the context sets $\mathcal{X}_\mathsf{S}$ and $\mathcal{X}_\mathsf{T}$ of two different problem instances with the same implicit relations.

**Intra-problem analogies.**   First, for each problem instance, we draw an analogy $\mathcal{X} :: \mathcal{X}'$ between a context set $\mathcal{X} \in \mathbb{R}^{8 \times h \times w}$ and its perturbation $\mathcal{X}' \in \mathbb{R}^{8 \times h \times w}$. The paired context set $\mathcal{X}'$ can be constructed by perturbing a random panel of the context set (for example, by replacing it with a noise image as in Figure 2a). Following the convention of existing meta-learning literature, we refer to $\mathcal{X}$ as the *support* $\mathcal{S}$ and its paired context set $\mathcal{X}'$ as the *query* $\mathcal{Q}$. We assume that $\mathcal{Q}$ is approximately analogical to $\mathcal{S}$ since its relation among the set elements remains unchanged even when removing a single element. By defining a *query* $\mathcal{Q}_i$ by replacing the $i$-th panel with a noise panel, we can prepare multiple analogical relations between the support and queries as follows:

$$\{\mathcal{S} :: \mathcal{Q}_1, \mathcal{S} :: \mathcal{Q}_2, ..., \mathcal{S} :: \mathcal{Q}_8\} =: \{\mathcal{S} :: \mathcal{Q}_i\}_{i=1}^{8}. \tag{1}$$

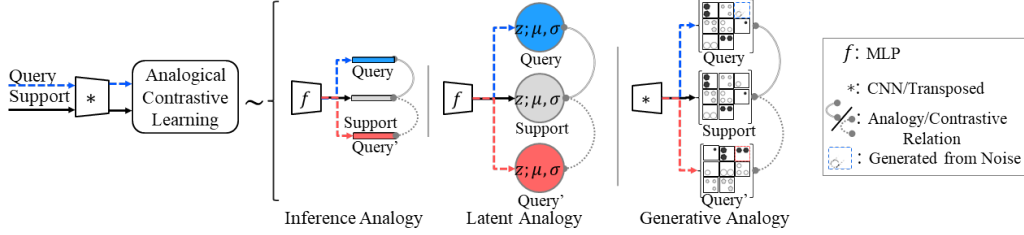

Figure 3: **Different types of analogical learning objectives.** 1) The **inference analogy** enforces the analogical problem instances to have similar output probability vectors, 2) the **latent analogy** enforce them to have similar latent representations, and 3) the **generative analogy** enforces the reconstructed panels to be similar.

Although this inter-problem analogy may look largely different from conventional notion of analogies, the two still constitutes an analogy as their relational structure is near-identical.

**Inter-problem analogies.** We can simply create an analogical pair of problem instances by generating two problems with the same semantics but with different attributes, as we know that the implicit relations between the elements in the two context sets will be the same even with the changes in the attributes (see Figure 2b). We refer to these instances from two different domains as source $S$ and the target $T$ respectively, where the $S \in \mathbb{R}^{8 \times h \times w}$ and $T \in \mathbb{R}^{8 \times h \times w}$. We can use the inter-problem analogies in our meta-analogical constrastive learning framework which we discuss later, where we learn by paring two semantically identical problems with randomly sampled attributes.

## 4.2 Analogical self-supervised contrastive learning

We now exploit the previously described analogies for relational representation learning. The basic idea is as follows: we maximize the similarity between a pair of analogical sets, while minimizing the similarity between a pair of non-analogical sets. To achieve this goal, we propose a simple self-supervised contrastive learning loss for each context set $\mathcal{X}$:

$$\ell_{\texttt{analogy}}(\mathcal{X}, \mathcal{X}_a, \mathcal{X}_n; \phi) := \log K_\phi(\mathcal{X}, \mathcal{X}_a) + \log\left(1 - K_\phi(\mathcal{X}, \mathcal{X}_n)\right) \qquad (2)$$

where $\mathcal{X}_a \in \mathcal{A}(\mathcal{X})$ is an analogical set to the given $\mathcal{X}$, $\mathcal{X}_n \in \mathcal{N}(\mathcal{X})$ is a non-analogical set, and $K_\phi(\cdot, \cdot)$ is a similarity kernel parameterized by $\phi$, with the value between 0 and 1. We define the $K_\phi(\mathcal{X}, \mathcal{X}')$ between the context set $\mathcal{X}$ and $\mathcal{X}'$ as $K_\phi(\mathcal{X}, \mathcal{X}') = \exp(-\lambda \cdot D(\mathcal{X}, \mathcal{X}'; \phi))$, where $D$ is a measure that defines the semantic distance between two relational embeddings parameterized by $\phi$, which we describe in detail in Section 4.2.1 and Section 4.2.2. We generate the set of non-analogical pairs $\mathcal{N}(\mathcal{X})$ by randomly perturbing the order of the elements (panels) in the original problem (see Figure 2c). Such a randomly perturbed context set could be considered as a hard-negative example, since although it contains exactly the same set of elements, it will not have the same relational structure with the original problem. Minimizing Eq. 2 will maximize the similarity between analogical pairs of sets, and minimize the similarity between the non-analogical pairs. Note that this is an unsupervised loss that does not require access to the ground-truth candidate set $\mathcal{C}$.

### 4.2.1 Intra-problem analogies

By modifying Eq. 2, we define the intra-analogy loss as follows:

$$\mathcal{L}_{\texttt{intra}}(\mathcal{S}; \phi) := \sum_{i=1}^{8} \log K_\phi(\mathcal{S}, \mathcal{Q}_i) + \sum_{i=1}^{8} \log\left(1 - K_\phi(\mathcal{S}, \mathcal{Q}_i')\right) \qquad (3)$$

where $\mathcal{S}$ is the support (original instance), $\mathcal{Q}_i$ is its intra-problem analogy generated by replacing a random panel with a noise panel, and $\mathcal{Q}_i'$ contains the same panels as $\mathcal{S}$ but with randomized ordering. We propose three different approaches to measure the distance between two context sets, which we describe in the following paragraphs.

**1) Inference analogy.** We first note that a hidden task $\mathcal{T}$ which generated the problem [3], should be the same across a pair of analogical context sets. Given a problem $\mathcal{S}$ and its paired problem $\mathcal{Q}$, we first encode each problem into the corresponding embeddings $\mathbf{s}$ and $\mathbf{q}$, using any existing neural networks (such as CNN, LSTM, CoPINet [3]). Then, by utilizing the task probability vector

$p(\mathcal{T}|\mathbf{s}) \in [0,1]^t$ ($t$ denotes the number of predefined number of rules) given a context embedding $\mathbf{s}$, we define the distance between the support $\mathcal{S}$ and the query $\mathcal{Q}$ as the binary cross-entropy as follows:

$$\mathcal{D}_{\mathtt{inf}}(\mathcal{S}, \mathcal{Q}; \phi) := -\mathbb{E}_{p_\phi(\mathcal{T}|\mathbf{s})}[\log p_\phi(\mathcal{T}|\mathbf{q})], \tag{4}$$

where inference probabilities $p_\phi(\mathcal{T}|\mathbf{s})$ and $p_\phi(\mathcal{T}|\mathbf{q})$ are modeled via the network parameterized by $\phi$ following by the sigmoid function. Between two analogical pairs, $\mathcal{D}_{\mathtt{inf}}$ should be close to zero.

**2) Latent analogy.** We further propose to learn the latent representation space of the relational structure on a certain (normal) distribution. We can measure distributional discrepancy between the latent variables of the two context sets as follows:

$$\mathcal{D}_{\mathtt{latent}}(\mathcal{S}, \mathcal{Q}; \phi) := KL[p_\phi(\mathbf{z}|\mathbf{s})||p_\phi(\mathbf{z}|\mathbf{q})] = -\mathbb{E}_{p_\phi(\mathbf{z}|\mathbf{s})}\left[\log \frac{p_\phi(\mathbf{z}|\mathbf{s})}{p_\phi(\mathbf{z}|\mathbf{q})}\right], \tag{5}$$

where $KL[\cdot||\cdot]$ denote the Kullback-Leibler divergence which is used to measure a probability distribution of query to that of support. We assume that random variable $\mathbf{z}$ is drawn from the multi-dimensional Gaussian distribution parameterized with the mean $\mu(\cdot)$ and the variance $\sigma(\cdot)$ which are modeled by neural networks, respectively (see Appendix B for further details).

**3) Generative analogy.** Given a pair of context panels $\mathcal{S}$ and $\mathcal{Q}$, we assume that one of the panels in $\mathcal{S}$ is replaced with a noise panel. If the pair constitutes an analogy, we should be able to reason about the content of the missing panel by inferring it from the matching panel from its analogical pair. This can be done by minimizing the distance between the original and the analogically reconstructed panel, which is defined as the negative likelihood as follows:

$$\mathcal{D}_{\mathtt{gen}}(\mathcal{S}, \mathcal{Q}; \phi) := -\mathbb{E}_{p_\phi(\mathcal{S}|\mathbf{s})}[\log p_\phi(\mathcal{S}|\mathbf{q})] \tag{6}$$

where $p_\phi(\mathcal{S}|\cdot)$ denote a conditional probability to estimate the original context panels which is modelled by a decoder network parametrized with $\phi$ following by the sigmoid function. The parameters of the decoder is learned with the additional loss, $-\mathbb{E}_{p_\phi(\mathcal{S})}[\log p_\phi(\mathcal{S}|\mathbf{s})] - \mathbb{E}_{p_\phi(\mathcal{S})}[\log p_\phi(\mathcal{S}|\mathbf{q})]$ where $p_\phi(\mathcal{S})$ is an empirical probability of the support image panels following by the sigmoid function.

### 4.2.2 Inter-problem analogy

We further consider the analogy across two different problem instances. If we know that the two problems have the same relations among the context panels, it will remain as the same problem even though their attributes (e.g., shapes) have been changed. We enforce an analogy between the two problem instances with different attributes (see Figure 2b). Similar to Eq. 4, we apply analogical contrastive learning between the instances from the source problem and target problem as follows:

$$\mathcal{L}_{\mathtt{inter}}(\mathcal{X}_{\mathsf{S}}, \mathcal{X}_{\mathsf{T}}; \phi) := \log\left(1 - K_\phi(\mathcal{X}_{\mathsf{S}}, \mathcal{X}_n)\right) + \begin{cases} \log K_\phi(\mathcal{X}_{\mathsf{S}}, \mathcal{X}_{\mathsf{T}}) & \text{if } \mathcal{X}_{\mathsf{S}} \text{ is analogical to } \mathcal{X}_{\mathsf{T}} \\ \log(1 - K_\phi(\mathcal{X}_{\mathsf{S}}, \mathcal{X}_{\mathsf{T}})) & \text{if } \mathcal{X}_{\mathsf{S}} \text{ is non-analogical to } \mathcal{X}_{\mathsf{T}} \end{cases} \tag{7}$$

where $\mathcal{X}_{\mathsf{S}}$ and $\mathcal{X}_{\mathsf{T}}$ are instances from the source and target domain $\mathsf{S}$ and $\mathsf{T}$, respectively, and $\mathcal{X}_n$ is the negative sample generated by shuffling the elements of the source context set $\mathcal{X}_{\mathsf{S}}$ (or $\mathcal{X}_{\mathsf{T}}$). The semantic distance between the two problems $\mathcal{X}_{\mathsf{S}}$ and $\mathcal{X}_{\mathsf{T}}$ are further defined as follows:

$$\mathcal{D}_{\mathtt{inter}}(\mathcal{X}_{\mathsf{S}}, \mathcal{X}_{\mathsf{T}}; \phi) := -\mathbb{E}_{p(r_\phi(\mathbf{e}_{\mathsf{S}}))}[\log p(r_\phi(\mathbf{e}_{\mathsf{T}}))] \times |\mathrm{d}(\mathbf{e}_{\mathsf{S}}, \mathbf{e}_{\mathsf{T}})| \tag{8}$$

where $\mathbf{e}_{\mathsf{S}} \in \{\mathbf{s}, \mathbf{q}\}_{\mathsf{S}}$ and $\mathbf{e}_{\mathsf{T}} \in \{\mathbf{s}, \mathbf{q}\}_{\mathsf{T}}$ denote the embeddings of $\mathcal{X}_{\mathsf{S}}$ and $\mathcal{X}_{\mathsf{T}}$ and $r$ denotes the relational embedding function parameterized with $\phi$. Instead of as hard binary categorization ($\{0, 1\}$) indicating positive or negative pairs in general contrastive metric learning [10], we use a soft similarity (analogy) metric between problems: $\mathrm{d} = -(\|p(\mathcal{T}|\mathbf{e}_{\mathsf{S}}) - p(\mathcal{T}|\mathbf{e}_{\mathsf{T}})\| - n)$ where $n$ is a constant to determine a positive/negative value. This metric is used to determine whether to minimize or maximize their distances, i.e., if $\mathrm{d} > 0$ (or $\mathrm{d} < 0$), $\mathcal{X}_{\mathsf{S}}$ is analogical (or non-analogical) to $\mathcal{X}_{\mathsf{T}}$.

### 4.3 Analogical contrastive learning for abstract visual reasoning

We now incorporate the previously defined analogy losses into the learning objective for visual reasoning tasks. To predict the correct answer panel ($y \in \mathbb{R}$) given a context set, we again use the

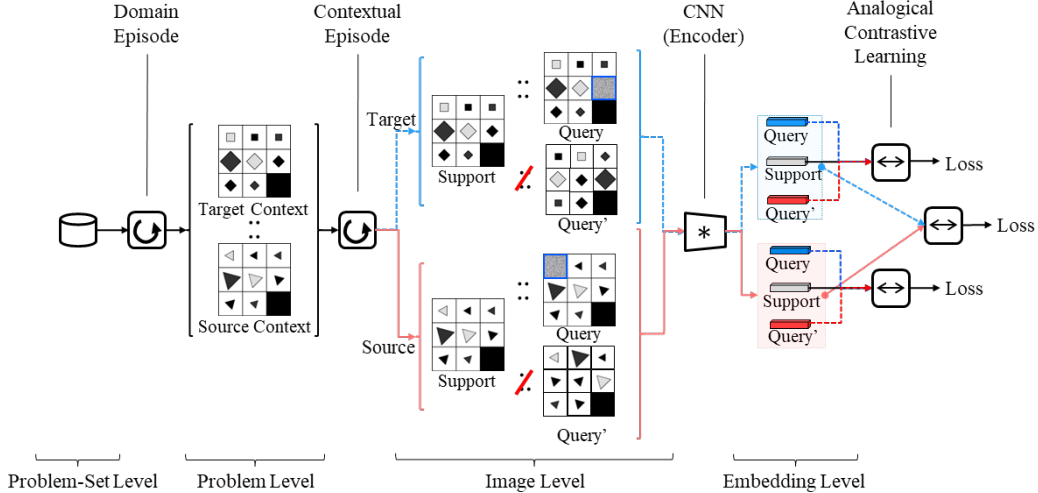

Figure 4: **Overview of our meta-analogical contrastive learning framework.** We maximize similarity between an instance and its intra-problem and inter-problem analogical pairs, while minimizing similarities between non-analogical pairs, over problems sampled with different attributes. This allows the model to learn relational representations that is agnostic to its surface appearance.

Noise Contrastive Estimation (NCE) learning objective to infer the final answer from a noise sample, following [3]. This framework has been used in many previous works related to detect contextual answers such as neural language modelling [18, 19], image captioning [11], and visual reasoning [3]. The NCE loss for all the panels, $\mathcal{L}_{\texttt{nce}}$ is given as follows:

$$\mathcal{L}_{\texttt{nce}}(\mathcal{X},\mathcal{C};\phi) := \ell_{\texttt{nce}}(\mathcal{S},\mathcal{C};\phi) + \sum_{i=1}^{8} \ell_{\texttt{nce}}(\mathcal{Q}_i,\mathcal{C};\phi), \tag{9}$$

where

$$\ell_{\texttt{nce}}(\mathcal{X},\mathcal{C};\phi) := \log\left(\sigma(\mathcal{X},\mathbf{c}_{y^*};\phi) - b\right) + \sum_{y\neq y^*} \log\left(1 - \sigma(\mathcal{X},\mathbf{c}_y;\phi) - b\right), \tag{10}$$

and $b$ is a scalar bias term [3]. In above, $\sigma(\cdot;\phi)$ is a neural network parameterized by $\phi$ with a sigmoid output and $y^*$ is the index of the correct answer. Intuitively, the NCE loss will maximize the probability of predicting the correct answer panel, while minimizing the probability of having an incorrect answer panel, by pushing the former to 1 and the latter to 0. The final combined loss for analogy reasoning for each problem instance $\mathcal{X}$ can be written as follows:

$$\mathcal{L}_{\texttt{instance}}(\mathcal{X},\mathcal{C};\phi) := \mathcal{L}_{\texttt{intra}}(\mathcal{X};\phi) + \mathcal{L}_{\texttt{nce}}(\mathcal{X},\mathcal{C};\phi). \tag{11}$$

For simplicity, we omit the balancing parameters for controlling the contribution of individual losses.

## 4.4 Meta-analogical contrastive learning

We now briefly describe our meta-analogical contrastive learning objective that includes the inter-problem loss $\mathcal{L}_{\texttt{inter}}(\mathcal{X}_{\mathsf{S}},\mathcal{X}_{\mathsf{T}};\phi)$ defined over samples with diverse pair of tasks that are defined by their attributes, such that it can generalize to unseen attributes. This is done by optimizing the final loss $\mathcal{L}_{\texttt{final}}$ over sampled tasks as follows:

$$\min_{\phi} \mathbb{E}_{\mathsf{S},\mathsf{T}\sim p(\mathcal{T})}\left[\mathbb{E}_{\mathcal{X}_{\mathsf{S}}\sim\mathsf{S},\mathcal{X}_{\mathsf{T}}\sim\mathsf{T}}\left[\mathcal{L}_{\texttt{inter}}(\mathcal{X}_{\mathsf{S}},\mathcal{X}_{\mathsf{T}};\phi) + \mathcal{L}_{\texttt{instance}}(\mathcal{X}_{\mathsf{S}},\mathcal{C}_{\mathsf{S}};\phi) + \mathcal{L}_{\texttt{instance}}(\mathcal{X}_{\mathsf{T}},\mathcal{C}_{\mathsf{T}};\phi)\right]\right]$$

$$(12)$$

where $\mathsf{S}$ and $\mathsf{T}$ are two tasks with different attributes sampled from the task distribution $p(\mathcal{T})$, $\mathcal{X}_{\mathsf{S}}$ and $\mathcal{X}_{\mathsf{T}}$ are sampled problems from each task, $\mathcal{C}_{\mathsf{S}}$ and $\mathcal{C}_{\mathsf{T}}$ are candidate sets, $\mathcal{L}_{\texttt{inter}}$ is the inter-analogy loss in Eq. 7, and $\phi$ is the model parameter. We illustrate an overview of our analogical contrastive learning framework in Figure 4.

# 5 Experiments

## 5.1 Experimental setup

We validate our method on a publicly available RPM dataset, RAVEN [2]. We resize the input images to $80\times80$ pixels, and train all models on NVIDIA Tesla V100 GPUs using the ADAM optimizer. We

Table 1: Accuracy (%) of the baselines and our analogical contrastive learning framework on the full RAVEN dataset. For baseline models, we report the accuracy of each model reported in [3]. L-R denotes the Left-Right configuration, U-D Up-Down, O-IC Out-InCenter, and O-IG Out-InGrid. We denote the best results for each task with bold fonts. AL and ACL denote our analogical self-supervised learning, and analogical constrastive learning, respectively. Batchsize=32.

| Method | Avg. Acc | Center | 2×2Grid | 3×3Grid | L-R | U-D | O-IC | O-IG |
|---|---|---|---|---|---|---|---|---|
| LSTM | 13.07 | 13.19 | 14.13 | 13.69 | 12.84 | 12.35 | 12.15 | 12.99 |
| CNN | 36.97 | 33.58 | 30.30 | 33.53 | 39.43 | 41.26 | 43.20 | 37.54 |
| WReN (Tag, Aux) | 33.97 | 58.38 | 38.89 | 37.70 | 21.58 | 19.74 | 38.84 | 22.57 |
| ResNet | 53.43 | 52.82 | 41.86 | 44.29 | 58.77 | 60.16 | 63.19 | 53.12 |
| ResNet+DRT | 59.56 | 58.08 | 46.53 | 50.40 | 65.82 | 67.11 | 69.09 | 60.11 |
| CoPINet | 91.42 | 95.05 | 77.45 | 78.85 | 99.10 | 99.65 | 98.50 | 91.35 |
| + AL (Ours) | 93.49 | **98.63** | 80.50 | 83.15 | **99.71** | **99.76** | **99.41** | 93.25 |
| + ACL (Ours) | **93.71** | 98.41 | **81.03** | **84.01** | 99.66 | **99.76** | 99.38 | **93.89** |
| Human | 84.41 | 95.45 | 81.82 | 79.55 | 86.36 | 81.81 | 86.36 | 81.81 |

set the number of epoch using the validation set provided in the dataset. As for the baselines, we use publicly available implementations to reproduce the results reported in [1, 3]. We provide more details of the experimental settings in the Section A of the Appendix. The RAVEN dataset consists of 70,000 problems, equally distributed across seven different tasks (e.g. 'Center', '2×2 Grid', '3×3 Grid', etc) [2, 3]. The dataset is split into ten subsets, where six of them are used as the training set, two are used for validation, and the remaining two are used for test. We compare our model against both simple baselines, namely (LSTM [20], CNN [21], vanilla ResNet [22]), and two strong baselines (WReN [1] and ResNet+DRT [2]) whose performances are reported in Zhang et al. [3].

## 5.2 Experimental Results

**Effect of analogical contrastive learning.** We first evaluate the generalization performance of the proposed method using the full training data, by comparing the test accuracy (%) of different models on the RAVEN dataset. We only use inference analogy as the intra-problem analogy for this experiment since latent and generative analogies require additional encoder and the decoder which are slow to train on large datasets. As shown in Table 1, our proposed analogical learning frameworks, AL (ACL without negative pairs) and ACL, yield significant performance improvement over existing methods. Moreover, ACL, which uses the contrastive loss with non-analogical pairs as the negative pairs, outperforms AL with only the positive analogical pairs, which demonstrates the effectiveness of the hard negative examples, that contain the same elements as the original problem with randomized ordering of elements. Although CoPINet surpasses human performance on this task, leaving little room for any improvement, our method obtains significant performance gains over it. Both WReN and ResNet+DRT, which mainly target PGM, heavily rely on additional supervisions such as rule specifications and structural annotations. However since the RAVEN dataset provides relatively little information (e.g. all ['Type', 'Size', 'Color'] parts are encoded as [1,1,1] among 9-digit rule-attributes tuple) compared to the PGM dataset, WReN obtains very low performance on RAVEN when using the additional supervisions. Thus we report the results of WReN without additional supervision. For a fair evaluation of the performance of all methods, we set all settings, including the batch size ($B = 32$) to be the same across all baselines and our model.

**Results on limited amount of training data.** The experimental results in the previous paragraph are obtained from the models trained on the full training set, which contains around 20,000 problem instances. However, as mentioned in the introduction, an important advantage of analogical reasoning is its ability to learn useful knowledge with a limited amount of samples. To validate this point, we further evaluate our model with extremely small number of samples (14 to 651), which correspond to 0.049 % or 1.6 % of all training images. We also evaluate our model only with analogical learning without negative pairs, to clearly demonstrate the effects of different types of analogical self-supervised learning. As shown in Table 2, all variants of our analogical self-supervised learning achieves significantly higher performance over the state-of-the-art method, CoPINet on this few-shot abstract reasoning task. We further observe that CoPINet+Analogies, which simply augments the original data with generated analogies, often obtain similar performance to the base CoPINet,

Table 2: Accuracy (%) of the baselines and our models with different analogical self-supervised learning approaches on the RAVEN dataset under few-shot settings. The full training set has $n = 42,000$ samples, and we divide the number of samples by $2^6$ (651) through $2^{10}$ (14). CoPINet+Analogies denotes the base model, CoPINet, trained with analogical examples generated using our method, but without the analogical contrastive learning. The bold fonts indicate the best accuracy for each sample size (row-wise). Batchsize=2.

| # samples | CoPINet | +Analogies | +Inference | +Latent | +Generative | +All |
|---|---|---|---|---|---|---|
| 14 (0.05%) | 24.29 | 28.12 | 28.12 | 27.55 | **30.42** | 30.37 |
| 35 (0.08%) | 32.45 | 34.62 | 36.65 | **41.79** | 36.24 | 33.55 |
| 77 (0.18%) | 45.24 | 45.57 | 45.56 | 46.02 | **51.22** | 47.04 |
| 161 (0.38%) | 50.00 | 50.55 | 53.35 | 51.41 | 53.37 | **56.97** |
| 322 (0.77%) | 54.25 | 58.09 | 59.42 | 59.28 | 60.50 | **64.26** |
| 651 (1.56%) | 60.02 | 69.11 | 67.23 | 68.13 | 65.64 | **71.76** |

Table 3: Accuracy of baselines and our methods (%) on few-shot RAVEN tasks with unseen attributes. The bold fonts indicate the best test accuracies. Batchsize=2.

| Method | Tasks | | | | | | |
|---|---|---|---|---|---|---|---|
| | Center | 2x2Grid | 3x3 Grid | L-R | U-D | O-IC | O-IG |
| CoPINet | 57.50 | 21.25 | 46.25 | 82.50 | 90.00 | 55.00 | 43.75 |
| +AL (Ours) | 75.00 | 31.50 | 46.25 | 82.50 | 90.00 | 63.75 | 50.00 |
| +ACL (Ours) | 76.25 | 32.50 | 46.25 | 86.25 | 90.00 | 66.25 | 53.75 |
| +Meta-AL (Ours) | 70.00 | **50.00** | 58.75 | 86.25 | 92.50 | **72.50** | 55.00 |
| +Meta-ACL (Ours) | **77.50** | 46.25 | **70.00** | **88.75** | **95.00** | 67.50 | **63.75** |

which shows the importance of enforcing explicit analogies via our proposed analogical contrastive learning. Also, Generative analogies outperformed other approaches when trained with extremely few examples (14-77) which makes sense since it provides more information to better guide the learning of the correct representation, via pixel-level supervision. However, with larger number of examples (161-651), using all types of analogies worked the best. For this experiment, we used a smaller minibatch size ($B = 2$), since we empirically observed that using a small batch size works better than with all methods, including the baseline CoPINet.

**Generalization to visual reasoning problems with unseen attributes.** To further validate the generalization performance of our model, we evaluate the performance of our model and CoPINet on test instances with unobserved attributes, from the RAVEN dataset. We use training samples that contain the same type of visual reasoning problems along a training set and validation/test sets but different shapes. Specifically, problem instances from the training set contains shapes such as ("triangle", "square", and "hexagon"), and problem instances in the validation/test set contains ("pentagon" and "circle"). We report the performance of our AL and ACL with the best-performing type of analogy (inference analogy, latent analogy, and generative analogy) for this experiment, selected using the RAVEN validation set. The results in Table 3 show that our proposed meta-analogy learning frameworks (Meta-AL and Meta-ACL) obtain significantly improved performance over both the base CoPINet and our models without meta-learning.

# 6 Conclusion

In this paper, we proposed a framework to learn the representations for abstract relations among the elements in a given context, by exploiting the *analogical relations* among them. Specifically, we focused on the analogy between the set representing the original problem instance and its perturbation, as well as to another problem instance from the same task class with different attributes, and then enforced the representations of the analogical pair of instances to be as similar as possible to discover implicit relational similarity between them, while minimizing similarities between non-analogical pairs that contain the same elements but in different layouts. Moreover, to allow the model to solve for unseen tasks, we further meta-learn our contrastive learning framework over pairs of samples with different attributes. While we mostly focused on a visual reasoning task in this work, our proposed analogical learning strategy may be applicable to other domains (e.g. natural language tasks) as long as there exists relational similarity between a pair of problem instances at any abstraction level.

## Statement of Broader Impact

We investigated a low-shot visual reasoning problem, which requires a higher-level relational reasoning skills that goes beyond perception of each individual elements. To this end, we introduce a new visual analogical learning framework that allowed the model to learn relational structure among the set elements from a pair of problems, which achieves significantly improved generalization performance with a limited amount of training samples, and generalizes well to problems with unseen attributes. Although we only considered a specific visual reasoning problem in this work, the proposed method is sufficiently general and we believe that the same analogical learning scheme could be applied to a wide variety of applications from non-vision domains, such as natural language understanding. Analogical reasoning is also a unique property of human intelligence and thus our work may be in the footsteps of the progress toward implementing artificial general intelligence (AGI).

## Disclosure of Funding

We received no third party funding for this work.

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
