[Supplementary Material]

# A  Experimental results

In this section, we examine our proposed meta-analogical contrastive learning using pairwise input samples from cross-domains. Firstly, we show an effect of contrastive learning based on relation between problems in addition to analogical learning. Then, we examine if our analogical learning is effective on analogical sampling based learning framework such as existing few-shot meta learning framework. Additionally, we show generalization performance of our proposed method across different visual domains.

**Analogy sampling based meta learning.**   In this section, we experimented analogy sampling based reasoning using the given RPM problem definition (called problem category or task, as mentioned above) from RPM dataset generation. The meta information from RPM dataset includes discrete codes composed of attributes and rules. With the given *problem category (task)*, a subset for learning can be sampled (via domain episode module in Figure 4 in main text). In each subset, a training sample and a test sample can have analogy relation dependent on their problem categories. In this experiment, we apply conventional meta (few-shot) learning (e.g. Model-Agnostic Meta-Learning (MAML)) [23], which consists of $K$-shot and $N$-class training samples for classification task. Here, by replacing *class* with *task*, $K$-shot and $N$-task reasoning framework can be defined. With this setting, we examined our proposed method using analogical sampling which contains different attributes (shape) but similar rule for meta-training (source) and meta-test (target) sets. This analogy sampling resembles hard task label based contrastive learning.

Here, we show analogical learning with the existing meta learning framework for fast adaptation from the source domain to the target domain. As similar to above experiments, CoPINet is used to baseline model in MAML. We use a public PyTorch based MAML code[1]. We experimented this method on the same dataset used in the experiment above while meta-test using training and test samples is different from test settings in the above. Experiments are conducted with settings: 1-shot training and 1-shot test, number of epochs = 200, batchsize = 16, and number of batches = 100 without hyperparameter optimization.

As shown in Table 4, our proposed analogy learning with analogy sampling (+Analogy Learning in the table) shows significant performance improvement in terms of test accuracy at the best validation accuracy compared to analogy sampling based meta learning (Analogy Sampling in the table). Hence, this result brought a new direction of analogy sampling based analogy learning via the meta-learning framework.

Table 4: Test accuracy (%) at the best validation accuracy along different number of adaptation ways (tasks). Different visual shape attributes are used among train and valid/test sets. The best accuracy is indicated using a bold font.

| number of ways | Analogy Sampling | +Analogy Learning (Ours) |
|:---:|:---:|:---:|
| 2 | 40.13 | 59.17 |
| 3 | 64.63 | 82.58 |
| 4 | 62.89 | 90.43 |
| 5 | 55.10 | 89.90 |
| 6 | 59.05 | **95.67** |

**Generalization across different configurations.**   We experimented our proposed method across different configurations on RAVEN dataset including: 2×2Grid ("distribute four", corresponding folder name in the dataset), 3×3Grid ("distribute nine"), O-IC ("in center single out center single"), O-IG ("in distribute four out center single"), L-R ("left center single right center single"), and U-D ("up center single down center single"). At each configuration, while shapes can be shared, overall visual arrangements are different each other. As similar to above experiment, we use training samples with three shapes ("triangle", "square", and "hexagon") and val/test sample with two different shapes ("pentagon" and "circle") to training for a generalization purpose.

Table 5: Test accuracy (%) at the best validation accuracy along different target configurations (domain) from "center" configuration to measure generalization performance on cross domain problems.

| Target Domain | Base (CoPINet) | Analogy (Ours) |
|:---:|:---:|:---:|
| 2×2Grid | 23.75 | 38.75 |
| 3×3Grid | 18.75 | 36.25 |
| O-IC | 30.00 | 31.25 |
| O-IG | 26.25 | 48.75 |
| L-R | 18.75 | 25.00 |
| U-D | 26.25 | 26.25 |

To show generalization reasoning performance of source domains on target domains, our model is trained using samples from "center" configuration (training) and then tested samples from other six configurations. As shown in Table 5, our proposed method ("+Analogy") shows significantly improved generalization performance on cross domain problems.

## B  Experimental settings

**Implementation.**  For reasoning experiments, we used PyTorch code based implementation to reproduce the baseline methods (LSTM [20], CNN [21], and vanilla ResNet [22]) and strong baselines (WReN [1], ResNet+DRT [2], and CoPINet) based on publicly available PyTorch code[2]. For dataset generation, we used publicly available RAVEN PyTorch code[3] in the experiment "Cross domains for unseen visual experience".

**Hyper-parameter settings.**  All models were trained using the Adam optimiser, with exponential decay rate parameters $\beta_1 = 0.9$, $\beta_2 = 0.999$, and $\epsilon = 1e-8$. We fixed the random seed ('12345', following open codes of the baseline method), and learning rate $1e-4$. An input size of image panels are resized to $80 \times 80$ from $160 \times 160$. Images are normalized into $[-1, 1]$. Any other transformation or data augmentation techniques are not used.

**Model details.**  Here, we provide details for all our models.

**1) Task inferences.** An architecture of the task inference encoder is defined in detail as follows,

$$e(\cdot) := [\mathsf{MLP}_1, \mathsf{reshape}, \mathsf{softmax}, \mathsf{MLP}_2, \mathsf{sum}]$$

where

- $\mathsf{MLP}$ consists of an inner-product layer, $\mathsf{MLP}_1 \in \mathbb{R}^{(d \times (a \cdot r))}$, $\mathsf{MLP}_2 \in \mathbb{R}^{(r \times 64)}$,
- $d$ denotes a dimensionality of a (feature) vector before a classifier,
- $a$ denotes the predefined number of attributes, $r$ number of rules, ($a = 10$, $r = 6$)
- $\mathsf{reshape}$ makes the vector size from ($\mathsf{batchsize} \times (a \cdot r)$) to ($\mathsf{batchsize} \cdot a \times r$),
- and $\mathsf{sum}$ make summation along $a$ to make a feature vector as ($\mathsf{batchsize} \times r$).

**2) Variational context latent.** We provide an architecture of the variational encoder in detail as follow,

$$v(\cdot) := [\mathsf{MLP}_\mu, \mathsf{MLP}_\sigma]$$

where $\mathsf{MLP}$ consists of an inner-product layer, $\mathsf{MLP}_\mu \in \mathbb{R}^{(d \times p)}$ for the mean latent vector, $\mathsf{MLP}_\sigma \in \mathbb{R}^{(d \times p)}$ for the standard-deviation latent vector, $d$ denotes a feature dimension before a classifier, and $p = 256$.

**3) Generative contextual images.** We provide an architecture of the denoising decoder for context images generation in detail.

$$d(\cdot) := [\mathsf{ConvT}_1, \mathsf{BN}, \mathsf{DeResCNN}_1, \mathsf{ConvT}_2, \mathsf{BN}, \mathsf{concat}([\mathsf{DeCNN}_1, \mathsf{DeCNN}_2]),$$

$$\mathsf{ConvT}_3, \mathsf{upsample}, \mathsf{reshape}]$$

where

- an input size is ($\mathsf{batchsize} \cdot \mathsf{channelsize} \times 64 \times 20 \times 20$),

- $\mathsf{ConvT}_1$ denotes a *ConvTranspose2d* layer (which is a transposed version a 2D convolutional layer implemented in PyTorch) parameterized with [input-channel: 64, output-channel: 128, kernesize: 3, stride: 2, padding: 1, bias: False] in PyTorch,

- $\mathsf{BN}$ denotes the batch normalization,

- ResNet block with transposed convludtional layers,

$$\mathsf{DeResCNN}_1 := [\mathsf{ConvT}_4, \mathsf{BN}, \mathsf{ReLU}, \mathsf{identity} + [\mathsf{ConvT}_5, \mathsf{BN}], \mathsf{ReLU}]$$

  with $\mathsf{ConvT}_4$ (*ConvTranspose2d*) with [input-channel 128, output-channel: 128, kernelsize: 3, stride: 1, padding: 1, bias: False], $\mathsf{ConvT}_5$ (*ConvTranspose2d*) with [input-channel: 128, output-channel: 128, kernelsize: 3, stride: 2, padding: 1, bias: False],

- $\mathsf{concat}$ denotes a concatenation,

- $\mathsf{ConvT}_3$ (*ConvTranspose2d*) with [input-channel: 64, output-channel: 1, kernelsize: 5, stride:2, padding:0, bias: False],

- $\mathsf{upsample}$ makes an output ($80 \times 80$) size,

- $\mathsf{reshape}$ makes ($\mathsf{batchsize} \cdot \mathsf{channelsize} \times 1 \times 80 \times 80$) to ($\mathsf{batchsize} \times \mathsf{channelsize} \times 80 \times 80$),

- A transposed convolutional layer for column-wise computation,

$$\mathsf{DeCNN}_1 := [\mathsf{ConvT}_6, \mathsf{BN}, \mathsf{ReLU}]$$

  with $\mathsf{ConvT}_6$ (*ConvTranspose2d*) with [input-channel: 64, output-channel: 32, kernesize: 3, stride: 2, padding: 0, bias: False], and

- A transposed convolutional layer for row-wise computation,

$$\mathsf{DeCNN}_2 := [\mathsf{ConvT}_7, \mathsf{BN}, \mathsf{ReLU}]$$

  with $\mathsf{ConvT}_7$ (*ConvTranspose2d*) with [input-channel: 64, output-channel: 32, kernesize: 3, stride: 2, padding: 0, bias: False].

## Footnotes

[1]available at https://github.com/tristandeleu/pytorch-maml

[2]available at https://github.com/Fen9/WReN and https://github.com/WellyZhang/CoPINet

[3]available at https://github.com/WellyZhang/RAVEN