[Reviews · NeurIPS 2020]

Review 1

Summary and Contributions: The paper proposes an efficient learning algorithm for the visual reasoning task of Raven's Progressive Matrices (RPM). Specifically, the authors propose a method inspired by human analogical reasoning and augment an existing model with several aspects of analogy, i.e., task inference analogy, variational context analogy, and generative analogy, implemented as auxiliary losses. In the experiments, the authors show that the proposed approach achieves better performance, learning efficiency, and generalization ability in the RAVEN dataset.

Strengths: + A good motivation. The work is motivated by human's analogical reasoning ability, which has not been adequately discussed and applied in the problem, despite evidence that RPM is well connected with the ability to understand analogy. + A unique aspect into the problem. Most existing computational works on RPM, especially recent ones that use deep learning to address the problem in a data-driven manner, do not consider learning efficiency, which I believe is important for AI to be widely available. + Good performance. The proposed method achieves very good performance in low-sample regime and demonstrates a certain level of generalizability.

Weaknesses: - Inaccurate advertising. While analogical reasoning is definitely a good motivating aspect for visual reasoning, the authors do not make it clear how their method is connected with analogy. From a technical point of view, the proposed losses are generative losses from three different angles, without any connection to analogy. The introduction discusses the four-word proportional analogy but fails to draw a clear picture of how the context panels in RPM problems naturally form cases of such analogy, making the method proposed disconnected from the motivation. - Some assumptions could be wrong and if they are not wrong, they should be clarified. 1. Line 85. The meaning of commutativity is very confusing. If the circle means composition and :: means equality, some RPM examples do not follow the general rules. From my understanding of the RPM generation method in [2], it only guarantees that the row swap OR the column swap will not break the hidden relation but if one performs both (AND), the relation will be broken. 2. The proportional analogy in Eq (1) is very confusing. I wonder when in the analogy pairs the heads are the same and tails different. 3. None of the proposed losses in Eq (2, 3, 4) relate to ``kernel''. They are neither symmetric nor positive semi-definite. 4. I wonder how in Eq (2, 3, 4) S and Q become random variables in distributions. From the supplementary material, they are just extracted features. And if they indeed are modeled as distributions, how are they modeled? 5. On the analogy losses. The so-called task inference analogy is quite unclear. From the formulation, it is maximizing the log likelihood, but how is that related to analogy? The same question can be asked on the variational context latent. The latents z do not look to preserve any semantics what does its KL measure? Still, how does maximizing joint likelihood in Eq (4) relate to analogy. Besides, S and Q are highly related, and if one assumes some analogy, they are certainly not independent. - Experiments to be completed. Another major dataset in RPM research is PGM. The authors mention it in related literature but do not run the proposed approach on it. Several regimes in PGM can certainly be good benchmarks for various aspects of generalization and efficiency. Showing similar performance on PGM will make the contribution more solid. *Post-rebuttal* I've read reviews from other reviewers and the authors' rebuttal. I agree with R2 that clarity is a serious issue for the paper and unfortunately, maybe due to the page limit, the rebuttal does not satisfactorily resolve it. For the additional experiments added in the rebuttal, it seems the improvement becomes marginal (only slightly better than random guess), compared to that reported in the main text. Therefore, I remain borderline.

Correctness: Most assumptions are but some may be problematic.

Clarity: There is much room for improvement. See details in weaknesses.

Relation to Prior Work: Yes.

Reproducibility: Yes

Additional Feedback:


Review 2

Summary and Contributions: The paper proposed a model that combined several forms of self-supervised learning to improve performance on RAVEN dataset, particularly when training data is scarse.

Strengths: 1. The proposed model shows slightly improved accuracy over CoPINet for both full and reduced training dataset. 2. The model shows some level of generalisation for unseen data.

Weaknesses: 1. The paper is not clearly written. This will be explained in the clarity section. 2. The proposed method seems just a combination of self-supervised learning ideas including VAE and NCE. Thus I don't think there is much novelty. 3. The authors only reported results on RAVEN dataset. However as noted in [1], the RAVEN dataset is biased and it is possible to use a cheating solution which can achieve good accuracy without looking at the context panels. [1] proposed balanced-RAVEN, which solves this problem. I would suggest authors to report results on this dataset. Moreover, there is another RPM dataset called PGM, on which results should be reported for this work to be considered a good model for RPM task. References: 1. Hu, Sheng, et al. "Hierarchical Rule Induction Network for Abstract Visual Reasoning." arXiv preprint arXiv:2002.06838 (2020).

Correctness: 1. Analogy via task inference. 'e(;\theta)' is said both to be an encoder and an inference network? I believe there is something wrong here. Moreover, it is not clear how p and q are computed. If they are just encoders, which output a vector, then log q(Q_i) is still a vector. How is this then turned into a scalar loss? 2. There are some notation overloading here. p and q are used several times for different things. It makes reading quite difficult.

Clarity: I have to say that this paper is not well written. There are grammar issues, overloading of notations, and unclear explanations (e.g. line 190 'domain mapping is found using a task category estimated based on rules...' It is not clear how task category is defined, and how is it estimated.) The authors just don't bother to explain these details, which make it hard to understand many parts of the paper.

Relation to Prior Work: Some references to prior works are included, but many others are missing, such as the one included in the weakness section. This paper does not have a 'related work' section, and I feel the relations to prior work is not properly discussed.

Reproducibility: Yes

Additional Feedback: 1. In equation 1, ':' and '::' is never defined, which makes it difficult to understand this equation. 2. Quite a few grammar issues. (e.g. line 159 to 160, ''Latent variables of embedding vectors can be measured their analogy as follows''). Maybe the authors should ask a native English speaker to proof-read. ------------------------------------ The authors addressed some of my concerns. However, in the authors' response they did not test on the full datasets of PGM and balanced-RAVEN. Thus it is not possible to tell if the methods perform better than previous SOTA works. For PGM only WReN, the simplest baseline, is compared against. Due to these issues I will keep my rating of 4.


Review 3

Summary and Contributions: The authors propose a method to bootstrap the training set in the RAVEN dataset, so that the learning method could extract the latent rule more efficiently. Experiments show a large performance improvement compared to the previous SOTA methods.

Strengths: + The proposed meta-analogical proposes a new direction to bootstrap the training set in order to more efficiently induce the latent rule in RPM problems. + The experiments show a large performance improvement over the SOTA method CoPINet. + Experiments are very thorough (in the main text and the supplementary material).

Weaknesses: The biggest concern is the fact that the authors seem to give too many criticisms to the recent papers on the RAVEN and RPM program in general. The central issue the authors raised is the "large" data used in recent methods, and the authors argue that the proposed method is "few-shot." Overall, the few-shot claim in this paper is considered valid, but the authors shouldn't treat the recent methods as if they use "large" data due to the following reasons. 1. The RAVEN dataset is overall in a small size since it's pixel input. PGM is slightly large, but still not sufficient to cover all dimensions in the pixels. After all, the intent of these two RPM datasets is indeed to tackle the small data learning problem. 2. The proposed bootstrapping method actually augments the additional data during learning. Although the data is not generated directly from the training data, it is still guided by the underlying rules to augment the training data. Hence, it's hard to say the proposed method is not using lots of data; perhaps, it is more precise to say the proposed method uses the training data more efficiently [L112]. However, one should not consider the efficient use of training data as opposed to using more data; these two do not always conflict. 3. Humans also rely on data. But what we are good at (compared to machines) is the ability to form abstraction and thus transfer to other problems. Noam Chomsky's argument of poverty of the stimulus is an indication along the line (though controversial). I feel the paper would be much more complete and properly framed in the literature if the authors could properly discuss the above points.

Correctness: See weakness.

Clarity: The paper is well-written and easy to follow.

Relation to Prior Work: The most related work CoPINet has been extensively discussed throughout the paper.

Reproducibility: Yes

Additional Feedback:


Review 4

Summary and Contributions: The paper introduces analogical reasoning, which aims to solve few-shot visual reasoning via similarity estimation. In particular, by modifying the context panels, authors propose contrastive learning algorithms that learn powerful representations for reasoning task. The experiments demonstrate non-trivial improvement over baselines.

Strengths: 1. The experiments seem to be sound and the improvements in accuracy and data efficiency are non-trivial. 2. Authors propose several learning strategies. 3. It is interesting to see that contrastive learning, or more generally, self-supervised learning can improve visual reasoning. 4. The code is provided to reproduce the results.

Weaknesses: 1. It is hard to understand the proposed approach after reading Section 3.2. For instance, in "task inference", what is the input space and output space of the inference network e? How are p and q modelled with the encoder? The next two methods are also hard to parse. It would be great if authors can clearly define and explain the terminologies. 2. The "Contrastive estimation" paragraph simplifies too many details. Equation (6) reveals no information about the NCE loss. For instance, [1] also use NCE loss for contrastive learning, but their equation (10) clearly define the NCE loss they use. 3. The dimensionality, input space, output space are not defined for the introduced variables and functions. [1] https://arxiv.org/pdf/1906.05849.pdf ----------------------------------------------- After viewing author's response, I still think the clarity can be improved. For instance, important notations and equations are inherited from previous works and some of them are left in appendix. Therefore, I will not change my score.

Correctness: The experiment setting is reasonable and sound. I would recommend the authors to clarify their method to improve correctness.

Clarity: Overall, I can understand the goal and the high level intuition. However, I have to read section 3 for several times to understand the methodology. Several notations are not well-defined (stated in Weaknesses) which makes the main approach hard to parse.

Relation to Prior Work: The main idea in this paper is contrastive learning, which is highly related to several self-supervised learning works for visual representations [1][2][3]. It would be great to have more discussion with these prior works. [1] https://arxiv.org/abs/1906.05849 [2] https://arxiv.org/abs/2002.05709 [3] https://arxiv.org/abs/1911.05722

Reproducibility: Yes

Additional Feedback: 1. Why is it called "Meta-analogical", in particular, what is the meta-training and meta-validation set? It seems that equation (7) only measures the similarity between source and target task? 2. Figure 2 is great. However, figure 3 and 4 are hard to parse. Simplifying them in a good way (removing the feature boxes) may improve clarity. They do not need to include every details.

[Author Response · NeurIPS 2020]

————————— **To Reviewers** R1, R2, R3, R4 —————————

Table A: Improved performances on PGM.

| #sams | Base | +Anal | +Anal+Inf | +Anal+Var | +Anal+Gen |
|---|---|---|---|---|---|
| 585 | 12.70 | 13.28 | 13.44 | 13.46 | **13.81** |
| 1171 | 13.37 | **13.91** | 13.90 | 13.49 | 13.59 |
| 4687 | 12.76 | 13.34 | 14.17 | **14.49** | **14.49** |

We thank all the reviewers for their insightful comments. In particular, we sincerely appreciate that reviewers (R1,R3,R4) acknowledge our proposed approach to be well-motivated, novel, and interesting. To improve our paper, we took the reviewers' advice to clarify and add details in our explanation for better understanding (R1, R2, R4), provide new insights to regarding the amount of data in visual reasoning problems (R3), and perform exhaustive experiments (R2). Finally, as suggested by (R1, R2), we provide experimental results on the PGM and Balanced-Raven dataset (see Table A,B).

(R1,R2,R4) **Details in Eq.(2,3,4) in Section 3.2.** We apologize for the oversimplified explanation. As mentioned in the manuscript, an analogy is learned by minimizing the difference between the (modelled) distributions over the objectives (*e.g.* Inference probability for task $\mathcal{T}$, $Pr(\mathcal{T}|S)$, hereafter $p(S)$) over transformed variables ($S$, embedding vector by CNN). Since the objectives have no ground truths, we use the estimated distribution from the support context as the reference. In short, we enforce the distributions over random variables which embed analogical relation to be similar. @R1 The distribution is modelled by a multivariate transformation (*e.g.* encoder). For empirical probability, a softmax function is followed by output of the transformation. We will include the terminology and notation in detail in the revision. @R2 A summation of these multidimensional BCE output returns a scalar value. $p(S)$ and $q(Q_i)$ indicate these empirical probabilities. @R4 Notations '$p$' and '$q$' simply follow the convention for BCE equation. We will include further explanations in the revision.

(R1, R2) **PGM.** As suggested, we experimented on PGM over a baseline (WReN) (Table A above), and the results show that our method outperforms it. We will include the results with more details in the revision.

(R1) **Connect the proposed method to analogy.** Our method include two analogies: $(a)$ within the context panels and $(b)$ between the context data (support/query). $(a)$ is explained with relational sets $\{\mathbf{x}_i : \mathbf{x}_j :: \mathbf{x}_r : \mathbf{x}_t\}$, where $\mathbf{x}$ denotes relational subsets ($\mathbf{x}$: combination of panels) of the context data $X$. $(b)$ is explained with $X$ and $\tilde{X}$ (perturbed $X$) and their embedding vectors $S$ and $Q$. Hence, we rephrase Eq.(1) with an analogy set $\mathsf{A} = \{S :: Q_i\}_{i=1}^8$. Analogy losses (maximum likelihood) assist to learn their analogies between modelled distributions. (R1) **Clarification.** ¶The four-word proportional analogy is merely a simple example and does not explain all the cases, as described in the manuscript. ¶L85: as noted in the paper, as the affine-transform based rule is a simple example but not a generalized one, we will eliminate it to avoid misleading. ¶Eq.(1): we will revise it to more direct expression ($\bigcup_i \{S :: Q_i\}$), which is also an analogy. ¶Kernel: we will replace this term with "similarity measure". ¶Analogy losses: as long as $S$ and $Q$ are highly related, analogy learning using probability models will work regardless of their independent relations.

(R2) **Argument on novelty.** Our novelty is the proposal of meta-analogical learning for efficient reasoning over limited observations. VAE and NCE, which R2 pointed out, are auxiliary or partial components to realize our approach. Our proposed framework is in a general scope so that it can be easily combined with the existing learning framework (*e.g.* MAML) as shown in the supplementary material. (R2) **Balanced-RAVEN.** Thank you for informing us about this dataset. Our method outperforms CoPINet on this dataset (Table B).

(R2) **More related works.** Most related works are explained in the **Preliminaries** section in L90-109. We will add a discussion with Hu *et al.* paper in the revision. (R2) **Clarification.** ¶Encoder and an inference network: as explained in L157, the task inference network uses the encoder. We will eliminate

Table B: Improved performances on Balanced-RAVEN.

| #sams | Base | +Anal | +Anal+Inf | +Anal+Var | +Anal+Gen |
|---|---|---|---|---|---|
| 14 | 14.77 | 15.71 | 15.51 | 14.81 | **15.11** |
| 35 | 15.70 | **17.17** | 14.81 | 16.42 | 15.52 |
| 77 | 15.57 | **15.93** | 15.81 | 15.96 | 14.74 |
| 322 | 16.66 | 16.75 | 17.11 | 16.37 | **17.31** |

L158 for better clarity. ¶':' and '::': even though it is explained in L33-35, we will add explanation (':' - relation and '::' - analogy) for clarity. ¶Grammar: we will proof-read it again. We will revise the notations as suggested.

(R3) **Criticisms and argument about data size.** Thank you for your insightful comments. 1) Although we were referring to the number of samples by the term "large", we agree that each sample has limited information since it contains small number of pixels, and will tone down on the strong arguments in [L2-3] and [L29-30]. 2) We will clarify that our method leverages the data in a more efficient manner as it is guided by the underlying rules. 3) We agree and believe that the analogy is one such example of such high-level abstraction, as it captures the relational similarities between inputs regardless of their raw input values.

(R4) **NCE.** As mentioned in the manuscript, NCE is a common learning objective and thus we did not emphasize it that much. We followed that of CoPINet: $\mathcal{L}_{\texttt{nce}} = \log\left(f(h_w(X, C(y)) - b)\right) + \Sigma_{y_i \neq y}\log\left(1 - (f(h_w(X, C(y_i)) - b))\right)$ where $b$ is a constant. We will further clarify this term in revision. (R4) **Meta-analogical?** For meta-learning of our model, we use the dummy-task labels using the predefined annotation (attributes and rules) as shown in the supplementary material (Table 2. in Section A.), and test on unseen types of attributes or rules at meta-test time. (R4) **Dimensionality.** This is given in the Section B of supplementary material. (R4) **Literatures.** The main idea of our work is not the use of contrastive learning for visual reasoning, but is the proposal of meta-analogical learning. Yet, we will include the suggested references in the revision. ¶Figure 3. and 4.: as suggested, we will simplify the figures in the final version.

[Meta-Review · NeurIPS 2020]

Reviewers were split on this paper, with two recommending accept and two recommending reject. Reviewers and the AC appreciated the strong performance and found the method to be intriguing if a bit complicated. A main concern was that the methods were hard to understand and the paper was not clearly written. The rebuttal addressed several confusions and partially satisfied the reviewers. The AC concurs that clarity is still an issue and urges the authors to revise the writing for the final version, especially in Section 3, making clear how each aspect of the method relates to the idea of analogies. See also the excellent suggestions on simplifying the diagrams (R4) and clarifying the technical details and notation (R1, R2), among other comments from the reviewers.